# Building Damage Visualization Through Three-Dimensional Reconstruction and Window Detection

**DOI:** 10.3390/s25102979

**Published:** 2025-05-08

**Authors:** Ittetsu Kuniyoshi, Itsuki Nagaike, Sachie Sato, Yue Bao

**Affiliations:** 1Division of Informatics, Graduate School of Integrative Science and Engineering, Tokyo City University, Tokyo 158-8557, Japan; 2Department of Architecture, Faculty of Architecture and Urban Design, Tokyo City University, Tokyo 158-8557, Japan; g2131086@tcu.ac.jp (I.N.); s-sato@tcu.ac.jp (S.S.); 3Department of Informatics, Yasuda Women’s University, Hiroshima 731-0153, Japan; bao@tcu.ac.jp

**Keywords:** architecture, remote sensing, 3D reconstruction, point cloud, sensor, visualization

## Abstract

This study proposes a non-contact method for assessing building inclination and damage by integrating 3D point cloud data with image recognition techniques. Conventional approaches, such as plumb bobs, require physical contact, posing safety risks and practical challenges, especially in densely built urban areas. The proposed method utilizes a 3D scanner to capture point cloud data and images, which are processed to extract building surfaces, detect inclination, and assess secondary structural components such as window frames. Experiments were conducted on prefabricated structures, detached houses, and dense residential areas to validate the method’s accuracy. Results show that the proposed approach achieved measurement accuracy comparable to or better than traditional methods, with an error reduction of approximately 19% in prefabricated structures and 21.72% in detached houses. Additionally, the method successfully identified window frame deformations, contributing to a comprehensive assessment of structural integrity. By applying gradient-based color mapping, damage severity was visualized intuitively. The findings demonstrate that this system can replace conventional measurement techniques, enabling safe, efficient, and large-scale post-disaster assessments. Future work will focus on enhancing point cloud interpolation and refining machine learning-based damage classification for broader applicability.

## 1. Introduction

### 1.1. Backgrounds

The causes of building damage are diverse, with natural disasters such as typhoons and earthquakes being particularly severe in terms of destruction. Rapid and safe damage assessment after disasters is crucial for preventing secondary damage and providing appropriate disaster relief. For instance, the Emergency Safety Assessment conducted after an earthquake and insurance damage assessments significantly impact decisions regarding building usability and insurance compensation [1]. These evaluations must cover a large number of buildings within a short period, yet current assessment methods rely on physical contact with structures, exposing inspectors to risks while also facing challenges related to measurement accuracy and manpower shortages.

Additionally, in Japan, many regions contain densely packed wooden houses [2,3], making physical access for measurement difficult in cases where buildings are closely situated. In wooden structure risk assessments, a phenomenon known as “Swing back” [4] can occur, where the maximum interstory drift angle is large, but the inclination measured during the survey appears small. This requires structural considerations unique to wooden buildings to ensure accurate evaluations.

Conventional assessment methods have various limitations. Traditionally, building evaluations involve direct physical contact, where inspectors use plumb lines and manual measuring tools to assess inclinations and damage. However, this method poses safety risks for inspectors, as they need to enter damaged structures. Moreover, results may vary based on the experience and judgment of individual inspectors, leading to a lack of consistency in evaluations. Furthermore, external factors such as wind and vibrations can affect measurements, leading to reduced accuracy.

Given these issues, there is an urgent need for a new non-contact building assessment technology that ensures accuracy, safety, and efficiency in post-disaster evaluations (Figure 1, Figure 2 and Figure 3).

### 1.2. Existing Technology

In recent years, 3D measurement technology utilizing point cloud data, which eliminates the need for physical contact, has gained significant attention. Hashikawa and Kanai et al. [5] proposed a method for automatically identifying and extracting standard steel sections from point cloud data obtained through the 3D laser scanning of steel bridge components. Similarly, Mishima and Kakizaki et al. [6] used 3D point cloud data measured with a laser profiler to detect residual deformations and damage in reinforced concrete after loading, employing plane detection techniques to quantify structural changes. However, their method is limited to localized wall measurements, making it unsuitable for evaluating damage to entire building facades.

Additionally, Xin and Asada et al. [7] utilized 3D point cloud data to detect edges on concrete wall surfaces, enabling damage assessment. However, since these methods rely exclusively on point cloud data, they lack image or video integration, which is essential for comprehensive damage verification. Meanwhile, Deng and Dou et al. [8] combined LiDAR data with aerial imagery to analyze terrain changes before and after earthquakes, allowing for the identification of specific damage locations and severity levels. However, because their method relies on aerial point cloud data, it is challenging to conduct high-resolution measurements at the individual-building level. Recent advancements have sought to integrate multiple sensing modalities to enhance damage assessment accuracy. For instance, Li et al. [9] introduced a multi-modal fusion method combining LiDAR and photogrammetry to improve post-disaster damage recognition, demonstrating superior performance in large-scale urban environments. Similarly, Kim et al. [10] proposed a deep learning-based segmentation model that integrates RGB and depth information from point clouds, enabling more detailed and accurate damage classification.

Moreover, studies such as Luo et al. [11] have explored automation in point cloud analysis. They developed a method that automatically extracts and classifies damage features from 3D point clouds using geometric and semantic data, greatly reducing manual annotation effort while maintaining high accuracy. However, their method depends on predefined damage categories, which may limit its applicability to novel or complex scenarios.

Consequently, while current approaches significantly contribute to the field of point cloud-based structural assessment, they often encounter challenges such as limited scalability, a dependence on prior damage models, and insufficient integration with visual inspection techniques. These limitations underscore the necessity for a more flexible and comprehensive approach—one that effectively combines both point cloud and image data for enhanced accuracy and adaptability in post-disaster assessment contexts.

Nevertheless, despite recent advancements, point cloud-based structural assessment techniques face critical challenges, including limited scalability, a dependence on predefined damage categories, and insufficient integration with visual inspection methods. Consequently, these approaches have yet to surpass the reliability and practicality of traditional plumb-line measurement, which remains the standard method for on-site structural evaluation.

## 2. Materials and Methods

In this study, we propose a method that utilizes a camera capable of simultaneously capturing 3D point cloud data and video to extract the entire building facade through point cloud processing and calculate its inclination. Furthermore, we evaluate the damage conditions of secondary structural components in densely populated residential areas based on wall inclination. This approach enables a comprehensive and detailed assessment of building damage, which has been challenging with traditional visual inspections and simple measurement tools. Additionally, we introduce a technique that directly integrates inclination levels into the building facade within video data, allowing for the seamless integration of wall assessments with image-based data.

### 2.1. Structure of the Proposed Method

Figure 4 presents the flowchart of the proposed method. Each step is explained in accordance with the flowchart below.

### 2.2. Building Imaging and Point Cloud Data Acquisition

A 3D scanner capable of capturing omnidirectional images is used for imaging and point cloud data scanning. The scan points are evenly distributed around the building, ensuring visual overlap between consecutive scan points. A specific light pattern is projected onto the target to enhance the accuracy of point cloud data acquisition. Table 1 presents an overview of the performance specifications of the 3D scanning cameras employed in this study.

The captured image and depth data are processed in the cloud, where the recorded images and point cloud data are integrated for further analysis. The image and depth data are processed using the Matterport cloud service, which performs automatic calibration and alignment. RGB images and 3D point cloud data are spatially aligned using the camera’s intrinsic parameters (such as focal length and principal point) and extrinsic parameters (pose and location at each capture point). This alignment enables a reliable correspondence between 2D visual features and 3D geometry. To validate registration accuracy, projected overlays of RGB images and point cloud data were manually reviewed. If visible misalignment exceeded 5 cm, a manual correction was performed based on reference features in both datasets. The 5 cm threshold was selected based on the minimum width of architectural components such as window frames, ensuring practical usability for structural evaluation.

### 2.3. Floor Removal and Clustering-Based Obstacle Removal

Point cloud data often contain unwanted objects unrelated to the building. Since the method focuses on measuring the inclination of wall surfaces, other planar surfaces, such as the ground, need to be removed. The ground removal process is carried out using RANSAC (Random Sample Consensus) [13], where the largest planar surface in the initial point cloud is identified as the floor and subsequently removed. Figure 5 shows the flowchart of this process.

Next, obstacles such as trees, utility poles, and vehicles introduce noise that affects window frame detection, making their removal necessary. To achieve this, a clustering algorithm is used to identify and eliminate small clusters that do not correspond to the building structure. Euclidean Cluster Extraction was used for this step. The minimum cluster size was set to 500 points, the maximum to 50,000 points, and the distance threshold between neighboring points to 0.1 m.

These values were selected based on empirical trials using various building types, including prefabricated structures and detached wooden houses. The 0.1 m distance threshold provided a good balance between preserving continuous wall geometry and removing scattered noise such as tree branches or vehicles. The minimum cluster size of 500 points ensured that small, irrelevant point groups (e.g., poles and foliage) were excluded, while the upper limit of 50,000 points was chosen to accommodate the full extent of typical wall surfaces or building facades. Figure 6 shows the flowchart.

Additionally, the scatter pattern, verticality, and linearity of the point cloud are analyzed using Principal Component Analysis (PCA) to enhance the distinct characteristics of various components. Figure 7 shows the flowchart of the process.

Figure 8 illustrates the results of these preprocessing steps.

### 2.4. Building Segmentation Using the Region-Growing Method

After removing obstacles, the region-growing method [14] is applied to segment each building individually. This ensures that measurements can be performed separately for each structure, even when multiple buildings are present in the dataset.

As shown in Figure 9, this method utilizes the normal vectors of the point cloud to cluster adjacent points based on the similarity of their angles. If the angle between the normal vectors of two points is below a certain threshold, they are considered part of the same structure. This allows building facades and structural components to be clearly segmented into distinct regions, enabling precise, object-wise analysis.

### 2.5. Wall Surface Extraction Using RANSAC

RANSAC (Random Sample Consensus) is used to extract wall surfaces from the segmented building point cloud. RANSAC is a robust plane detection algorithm that effectively identifies walls and flat surfaces while filtering out noise and outliers. Figure 7 illustrates the principle of RANSAC.

The process begins by randomly selecting three points to estimate a plane model. The distance between all data points and the estimated plane is then calculated. Points within a predefined threshold are classified as inlier points, while those beyond the threshold are considered outlier points. The model is then refined using only the inliers.

This process is iterated multiple times, with the model containing the largest number of inliers being selected as the final wall surface. In Figure 10, the results of the RANSAC model are compared with those of a traditional regression model, demonstrating its superiority in detecting structural surfaces.

### 2.6. Corner Extraction Using Convex Hull

To extract the plane at the building corners, which are the inclination measurement points, a convex hull [16] and point cloud interpolation are applied to the point cloud data extracted in Section 3.1.2. Figure 11 illustrates this process.

Since the convex hull represents the outermost shape of the point cloud data, it is used to extract the corner regions of the wall surface. When constructing the convex hull from the building’s point cloud data, the minimum and maximum points are utilized to define the building corners. As a result, key corner points are generated, and interpolation along the Z-axis is performed to refine the extracted points.

### 2.7. Point Cloud 2D Projection

As shown in Figure 12, projection images, depth images, and coordinate data are extracted relative to the building’s wall surfaces. These projection images from multiple directions are later used for matching with camera images, as described in subsequent sections.

### 2.8. Window Frame Detection Using 2D Camera Images

In this process, the extracted wall point cloud data are treated as an image, and machine learning is used to identify the window frames. Since existing models designed for point cloud data have certain limitations, RGB images captured by the Matterport camera are used instead. Figure 13 illustrates the architecture of the Window Detection iFacades Using Heatmaps Fusion [17] model, which is utilized for this task.

### 2.9. Creation of Frontalized Images Using Projection Transformation

Projection transformation is applied to the images acquired from the camera to frontalize the detected window frame images. This transformation ensures that inference is performed based on trained data, facilitating accurate window detection and easier matching with the projected point cloud images. Figure 14 illustrates the differences in window detection accuracy before and after frontalization.

### 2.10. Matching and Back-Projection of Window Frame Detection Results

A coordinate transformation is performed between the camera image and the projected point cloud image to align their positions. Based on the matching results, the detected window frame locations are back-projected onto the 3D point cloud data.

This process accurately determines the window frame locations in the original 3D space and extracts only the point cloud data corresponding to the window frames.

### 2.11. Extraction of Window Sashes and Calculation of Intersection Angles

Once the window frame point cloud data have been identified, the next step is to extract the window sash. The sash is part of the window frame and typically has a linear structure.

As shown in Figure 15, the normal vector of the approximated plane is calculated. The angle between this normal vector and the horizontal plane vector is then computed using Equation (1) to determine the wall inclination. In Equation (1), *a*, *b*, and *c* represent the x, y, and z components of the plane’s normal vector.(1)$$\theta =\arccos\left(\frac{b}{\sqrt{a^{2}+b^{2}}}\right)$$

### 2.12. Visualization of Risk Levels

To visually represent the inclination degree on the building’s walls within the video, the extracted plane point cloud is color-coded. This enables an intuitive visualization of the risk level associated with the building’s inclination.

For risk visualization, the hue and angle correspond to the hue (H) in the HSV color space and the wall inclination angle, respectively. A linear transformation is applied to map the specific inclination ranges to different hues.

As shown in Figure 16, the relationship between color and risk level is determined based on the Manual for Emergency Risk Assessment of Earthquake-Damaged Buildings [1]. The hazard level classification is defined in a risk assessment table based on inclination angles.

## 3. Results

To verify the effectiveness of the proposed method, the following four experiments were conducted:The accuracy evaluation of wall inclination measurement using a simple prefabricated structure and residential buildings.Risk assessment for actual residential and wooden structures.The detection of secondary structural components using full-scale wooden models and 3D models.The detection of window frames as secondary components in densely populated residential areas.

### 3.1. Experiment 1

To confirm the basic effectiveness of the proposed method, a small storage shed was selected as a simplified architectural model. The accuracy of wall inclination detection was evaluated by comparing it with the conventional plumb-line method, which is currently used in field surveys. A digital angle meter was used to obtain reference values, and the inclination measured by the proposed method was compared with that obtained using conventional method.

Additionally, to verify the applicability of the proposed method to actual residential buildings, measurements were conducted on a detached house in a residential area, followed by a precision comparison with conventional method.

#### 3.1.1. Field Measurement Method

As shown in Figure 17, the experiment was conducted using the following:

A storage shed with a depth of approximately 2.0 m, a width of 5.0 m, and a height of 2.8 m.

A two-story detached house with a depth of approximately 9.3 m, a width of 8.3 m, and a height of 5.4 m (shown in Figure 17).

To ensure the acquisition of comprehensive data, images were taken around the structure from a distance of approximately 2 m, allowing for the measurement of inclination at eight corner points (points 1 to 8) as shown in Figure 17.

The flow from point cloud data acquisition to corner extraction is illustrated in Figure 18.

#### 3.1.2. Experimental Results

The extracted corner point cloud and the color-coded results based on inclination are shown in Figure 19.

Since corner point clouds are prone to external infrared interference and matching errors, they often exhibit irregularities and surface unevenness. To achieve accurate inclination measurements, it is crucial to apply appropriate filtering and smoothing techniques to minimize measurement errors.

In this study, four commonly used filtering and smoothing methods were applied and compared for inclination measurement accuracy: curvature filtering, MLS (Moving Least Squares) smoothing [18], SOR (Statistical Outlier Removal) filter, and a Gaussian filter.

For processing (a), as shown in Figure 20, (b) and (c) exhibit residual unnecessary edge points, while in (d), the protrusions at the edges are suppressed, but distortions appear on the wall surface. Each method was evaluated based on normal deviation, inclination measurement error, and surface smoothness (standard deviation of curvature). The results are summarized in Table 2.The evaluation results confirmed that applying curvature filtering + MLS smoothing resulted in the smallest inclination measurement error, enabling an accurate estimation of the inclination angle. This method first removes localized outliers using curvature filtering and then applies MLS smoothing, effectively suppressing the influence of local noise while reconstructing a smooth and stable plane. As a result, the accuracy of normal estimation improved, and the inclination measurement error was minimized.On the other hand, the Gaussian filter enhanced surface smoothness, but it tended to blur edge features, making it less effective than MLS in reducing the inclination measurement error. The SOR filter was effective in removing outliers but could not completely suppress local variations in the point cloud, leading to higher inclination measurement errors.Based on these results, it was demonstrated that curvature filtering + MLS smoothing is the most effective method for the inclination measurement of corner point clouds, as it provides a well-balanced preprocessing approach that optimally combines noise removal and surface smoothness.Table 3 and Table 4 summarize the accuracy of the conventional and proposed methods, along with graphs shown in Figure 21. The metrics where the proposed method outperforms the conventional method are highlighted in red. The reference inclination angles used for accuracy evaluation were obtained by direct measurement using a digital angle meter. These values are referred to as “Measured Reference Data” in this paper. It is recognized that these measurements inherently include some degree of error or uncertainty. However, they were used as baseline values for comparing the inclination extraction results obtained via the plumb-line method and the proposed 3D point cloud processing method.

#### 3.1.3. Consideration

In the prefabricated structure experiment, regarding the relative error with respect to the reference values, the proposed method achieved smaller errors than the plumb-line method at five measurement points (2–5, 7). However, at three measurement points (1, 6, 8), the proposed method resulted in larger errors compared to the plumb-line method. Notably, at measurement point 8, the proposed method had the largest measurement error of 0.0597°, but this was still smaller than the maximum error of 0.0737° observed in the plumb-line method, demonstrating higher measurement accuracy.

Regarding the standard deviation, the proposed method exhibited a smaller standard deviation at five out of the eight measurement points (1, 3–5, 7) compared to the plumb-line method. At the remaining three points (2, 6, 8), the proposed method’s standard deviation was larger than that of the plumb-line method. However, at measurement point 6, where the proposed method showed the largest standard deviation (0.0692°), this value still remained below the plumb-line method’s highest standard deviation (0.0704° at measurement point 6). These results indicate that the reconstruction process enabled accurate plane detection and measurement despite matching errors or changes in point cloud shape caused by obstacles at edge locations.

In the residential building experiment, the proposed method resulted in smaller relative errors than the plumb-line method at five measurement points (1, 3, 6–8), while at three points (2, 4, 5), the proposed method’s error was larger. The largest measurement error of 0.123° occurred at measurement point 4 using the proposed method, which was 0.021° lower than the 0.102° error observed with the plumb-line method at the same location. However, at measurement point 7, where the plumb-line method had its maximum error of 0.124°, the proposed method recorded 0.001° higher at 0.125°, making it comparable to the plumb-line method in terms of accuracy. Additionally, at measurement point 3, the plumb-line method had an error of 0.1°, whereas the proposed method achieved a significantly smaller error of 0.001°.

Overall, when comparing all eight measurement points, the proposed method and the plumb-line method exhibited similar levels of variation in measurements, but the proposed method demonstrated more stable results and higher measurement accuracy.

### 3.2. Experiment 2

Since conducting on-site measurements of buildings damaged by actual earthquakes is challenging, we verified the presence of tilts in the buildings within the Gotoh Museum, which contains many wooden structures, and compared the results with those obtained using the plumb-line method.

#### 3.2.1. Field Measurement Method

Following the same procedure as Section 3.1.1, we captured images of the entire building and measured the inclination of the wall corner sections using both the plumb-line method and the proposed method (Figure 22).

#### 3.2.2. Experimental Results

The measurements confirmed that the Azumaya (a traditional Japanese pavilion) had actual tilts, and the results are presented in Table 5 and Figure 23. Items with higher accuracy are highlighted in red, while those with an inclination exceeding the reference threshold are underlined.

#### 3.2.3. Consideration

Among the seven measurement locations in the Azumaya, the proposed method showed smaller measurement errors than the conventional method at four locations (1, 2, 4, 5). The tilts detected using the conventional method were also successfully identified by the proposed method, and the proposed method achieved higher accuracy while also providing hazard-based color coding. These results suggest that the proposed method is capable of evaluating actual tilts in building walls, demonstrating its applicability for real-world structural assessments.

### 3.3. Experiment 3

In this experiment, we evaluated the accuracy of the proposed method by using an artificial model that replicated angled window sashes made of wooden materials. A digital protractor was used to measure the actual angles, which served as the reference values. Additionally, to compensate for the limited variation in real-scale models, a 3D model of a deformed building with secondary structural deformations was used for detection and evaluation.

#### 3.3.1. Experimental Method

Considering the effects of wooden house rebound and secondary structural deformations, we created window sash models (referred to as real-scale models) based on standardized dimensions [19] (Width (W) × Height (H) = 1690 × 1830 mm and 780 × 970 mm). The models were inclined within the plane to simulate real-world deformations. However, construction errors [20] associated with sash dimensions were negligibly small and thus considered insignificant.

The deformation angles were set according to the Emergency Safety Assessment criteria [1], specifically 1/60 rad (=0.954°) and 1/20 rad (=2.864°), which were used as the reference values.

#### 3.3.2. Experimental Results

The experimental results are presented in Figure 24 and Table 6.

For each window frame model, the window frame was successfully detected within the point cloud, and the intersection angle was calculated through linear detection.

In addition to the horizontal direction, deformations in the depth direction were also confirmed.

In Figure 25, blue represents areas with no inclination, while regions with inclination gradually shift to red, illustrating the tilt variation across the surface.

#### 3.3.3. Consideration

The experiments confirmed that the real-scale measurements of window sashes of various sizes could be performed using point cloud data. The results closely matched the target angles (1/20 rad and 1/60 rad), and the Mean Percentage Error (MPE) ranged from 0.442% to 2.875%, demonstrating sufficient measurement accuracy for this verification experiment.

By analyzing not only the horizontal plane but also the deformation in the depth direction, the impact of earthquakes can be assessed in greater detail. The overall inclination of the wall surface was visualized using a gradient representation, and the method successfully identified deformations and the detachment of secondary structural components, such as window frames. Through proper data acquisition and extraction methods, the system was able to detect not only tilting and detachment but also various deformation patterns, including bending and cracking.

Based on the results from both models, it was confirmed that the deformation of the building and the deformation of the secondary structural components can be accurately detected in point cloud data.

### 3.4. Experiment 4

This experiment aims to detect secondary structural components in a densely built residential area.

#### 3.4.1. Experimental Method

As shown in Figure 26, images and point cloud data were obtained by capturing a real residential area. The shooting distance and interval were kept constant, and data were collected over a distance of approximately 45 m. The objective was to extract window frame point clouds, which was achieved by detecting window frame regions from image data, matching them with projected point cloud images, and performing inverse projection to extract the corresponding window frame regions in the point cloud.

#### 3.4.2. Experimental Results

Figure 27 illustrates the segmentation results of individual buildings within a dense residential area.

As shown in Figure 27a, two green-colored buildings were erroneously clustered together. Additionally, the purple-colored building was grouped with a hedge and a utility pole in front of it, forming a single cluster. Similarly, the pink-colored building on the left was merged with an adjacent exterior fence. However, after applying Principal Component Analysis (PCA) filtering and clustering processing, the two green-colored buildings were successfully separated, as confirmed in Figure 27c.

Figure 28 shows the successful extraction of window frames from the point cloud.

#### 3.4.3. Consideration

As shown in Figure 27 and Figure 28, the results demonstrate that, even in densely built residential areas, the proposed method enables the segmentation of individual buildings within the point cloud, despite the presence of obstacles. Additionally, secondary structural components could be captured in images and extracted as window frames in the point cloud.

By capturing the buildings from a distance in a single shot, the proposed method reduces the time required for hazard assessment. Furthermore, since the data processing can be conducted in a safe environment, the safety of the assessment process is significantly improved.

In urban environments, obstacles such as street trees, fences, parked vehicles, and utility poles were found to partially obstruct building facades, resulting in missing point cloud data in those areas. To address this challenge, we propose two practical approaches for future implementation.

First, we can apply point cloud interpolation techniques such as Poisson Surface Reconstruction and Moving Least Squares (MLS) to geometrically estimate missing regions. In particular, MLS is effective in preserving local surface continuity and normal smoothness, which is ideal for restoring planar building surfaces.

Second, we suggest performing multi-view data acquisition from multiple angles (e.g., frontal, lateral, and oblique views). In our preliminary tests, capturing a building from at least three different viewpoints enabled the coverage of over 90% of its external surfaces.

Furthermore, combining fixed-position scanners with mobile LiDAR (e.g., handheld or drone-mounted) could provide additional coverage and flexibility for difficult-to-reach or highly occluded areas.

## 4. Discussion

The proposed method integrates 3D point cloud data and image recognition technology to provide a non-contact, efficient assessment of building damage. Through multiple experiments, it was confirmed that the proposed approach achieves comparable or superior accuracy to traditional measurement methods such as plumb bobs and protractors. Notably, in environments with high building density, where physical measurements are challenging, the proposed method demonstrated practical applicability.

Compared to conventional techniques, this method offers several advantages. First, it enables the evaluation of building inclination through image and video analysis, rather than relying solely on numerical measurements obtained from plumb bobs or protractors. Second, it was confirmed that the method achieves equivalent or superior measurement accuracy even for actual residential buildings. Third, the approach was validated in densely populated wooden housing areas, demonstrating its practical utility. Fourth, the detection of window frames in a three-dimensional space and the extraction of deformations in secondary components provided valuable insights into building damage assessment, particularly considering seismic effects such as structural rebound. Fifth, the use of a camera-based hazard assessment system allowed for the remote, non-contact measurement of entire buildings, thereby enhancing the safety of inspectors.

Despite these advantages, some limitations were observed. For instance, in residential areas with dense vegetation or obstacles such as fences and utility poles, the occlusion of building surfaces resulted in incomplete point cloud data. In such cases, supplementary data reconstruction techniques are necessary to infer missing regions. Additionally, variations in camera quality and LiDAR accuracy can impact point cloud resolution, which in turn affects the accuracy of inclination measurements. Future research should focus on optimizing point cloud interpolation methods and refining machine learning algorithms for enhanced structural damage assessment. For example, interpolation methods such as Poisson Surface Reconstruction and MLS can be integrated to recover missing geometry caused by occlusion. These methods are particularly effective for filling holes on planar or gently curved surfaces, which are common in building facades.

In addition, future work could explore combining data from oblique and alternative viewpoints to improve the completeness of facade reconstruction. Incorporating supplementary sensors—such as drones, handheld LiDAR, or smartphone depth sensors—may provide richer and more complete spatial data, especially in areas where traditional ground-level capture is insufficient.

On the algorithmic side, refining machine learning techniques for damage detection under partial occlusion and noisy conditions would further increase the robustness of the proposed system in real-world post-disaster environments.

## 5. Conclusions

This study proposed a novel building hazard assessment method by integrating 3D point cloud processing with image recognition. Through experiments conducted on prefabricated structures, detached houses, and densely built wooden residential areas, it was confirmed that the proposed method is effective in detecting inclination and assessing damage to secondary components.

The accuracy of the proposed method was validated by demonstrating measurement precision comparable to or exceeding that of conventional plumb bob techniques, even in practical field applications. Additionally, by integrating 3D point cloud analysis and image recognition, this method enabled hazard assessment without direct physical contact, enhancing both safety and efficiency. Another key advantage of this approach is its ability to account for post-seismic structural rebound effects, which often lead to underestimated inclination measurements in conventional methods. Furthermore, by detecting deformations in window frames and other secondary structural elements, this method provided a more comprehensive evaluation of structural integrity. Lastly, its applicability was confirmed in densely populated urban environments, highlighting its potential for large-scale post-disaster assessments.

These findings suggest that the proposed method offers a significant improvement over conventional approaches in terms of accuracy, safety, and scalability, making it a valuable tool for future building hazard assessments.

## Figures and Tables

**Figure 1 sensors-25-02979-f001:**
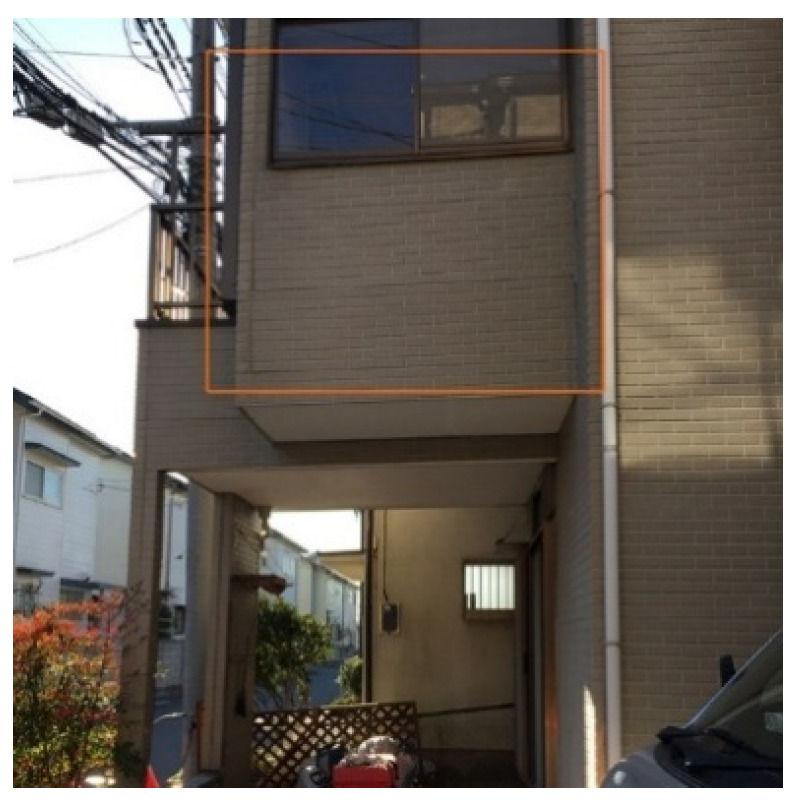
Wall inclination.

**Figure 2 sensors-25-02979-f002:**
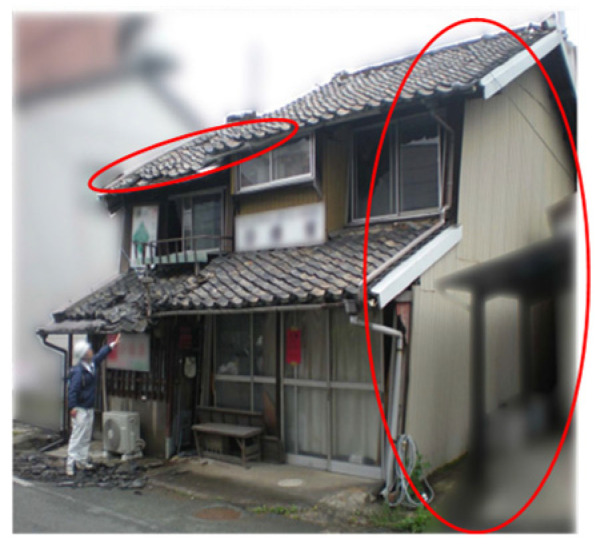
Damage to secondary structural components.

**Figure 3 sensors-25-02979-f003:**
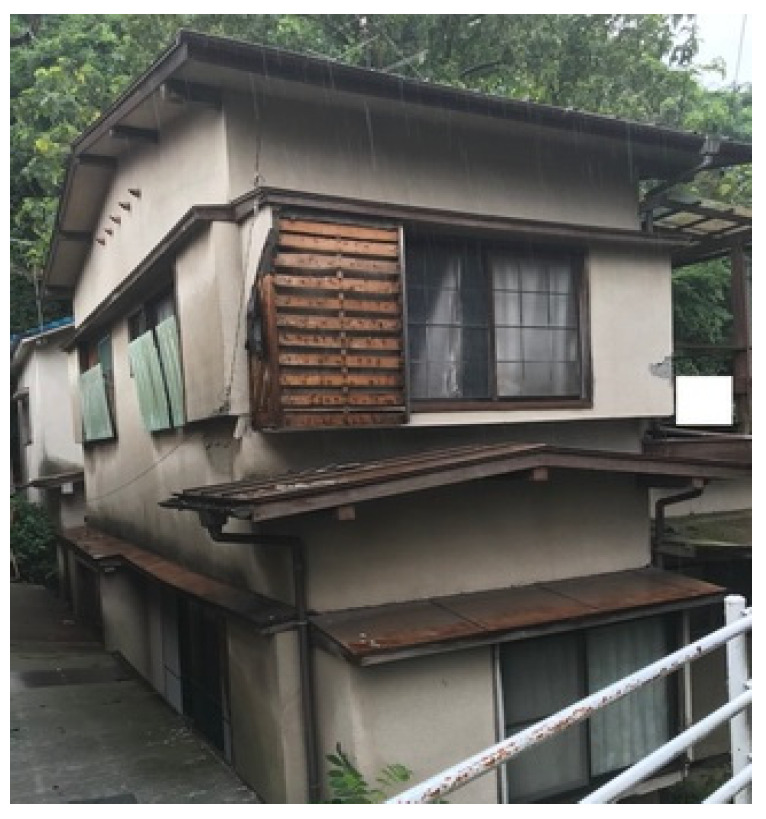
Overall structural damage.

**Figure 4 sensors-25-02979-f004:**
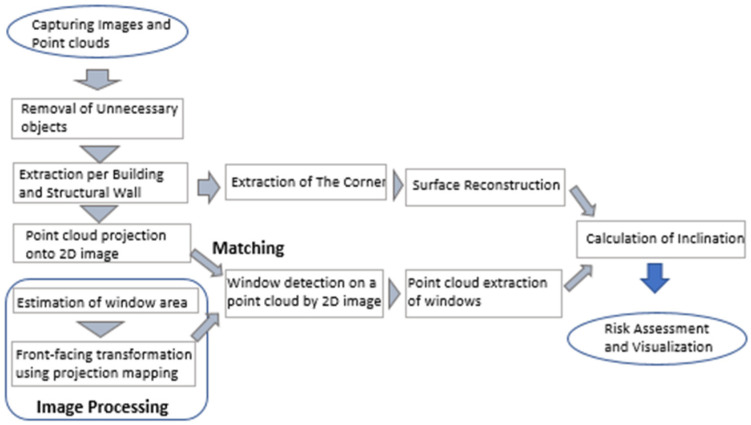
Flowchart of the proposed method.

**Figure 5 sensors-25-02979-f005:**
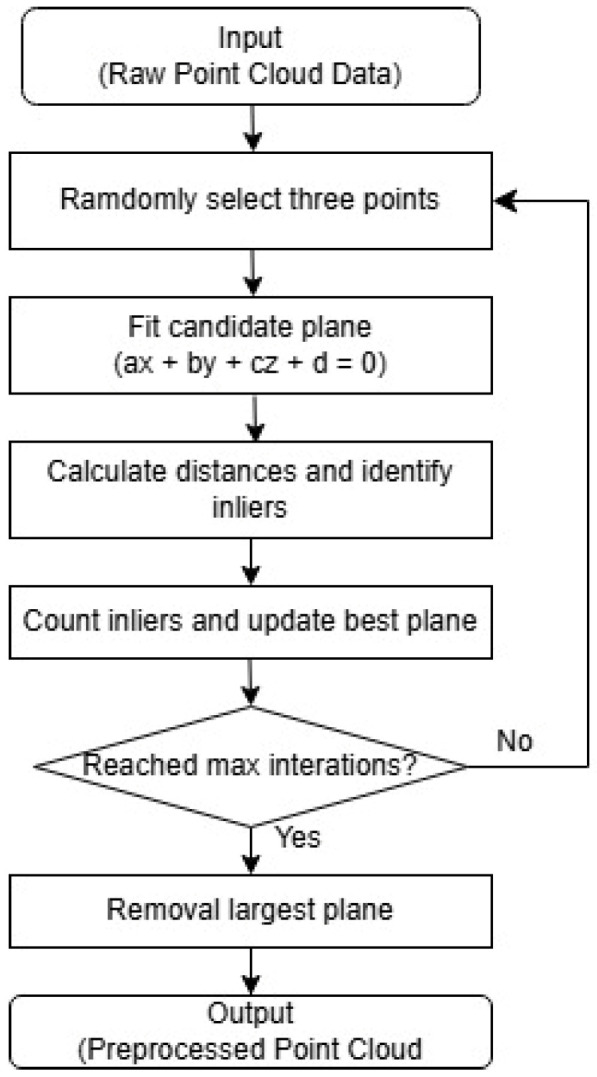
The RANSAC algorithm for floor removal.

**Figure 6 sensors-25-02979-f006:**
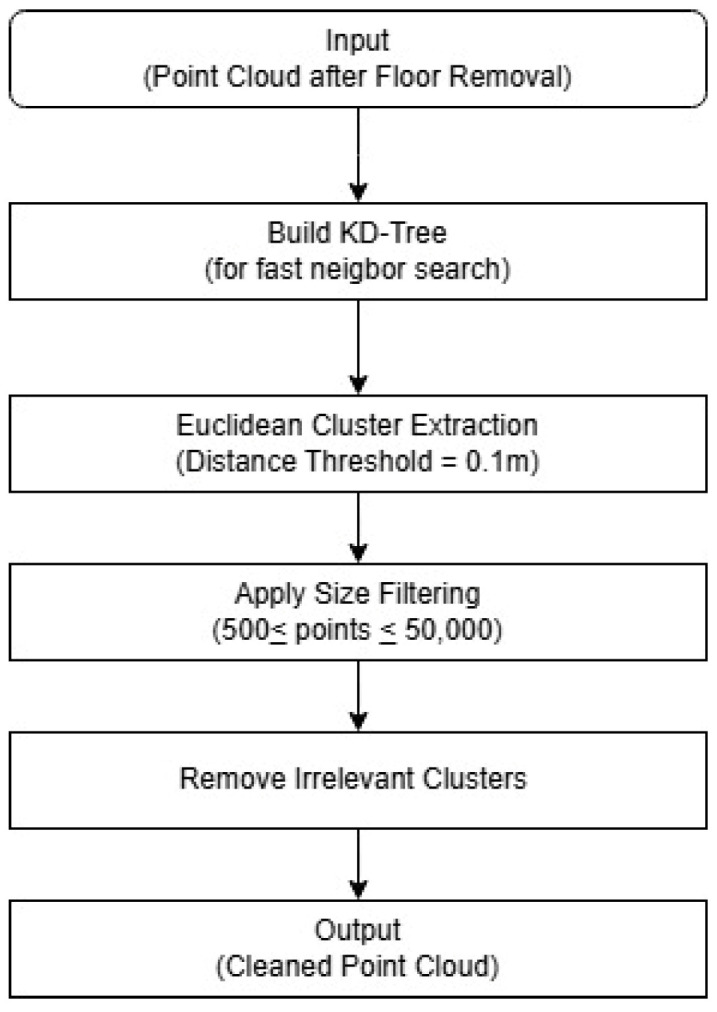
Flowchart of clustering-based obstacle removal (using Euclidean cluster extraction).

**Figure 7 sensors-25-02979-f007:**
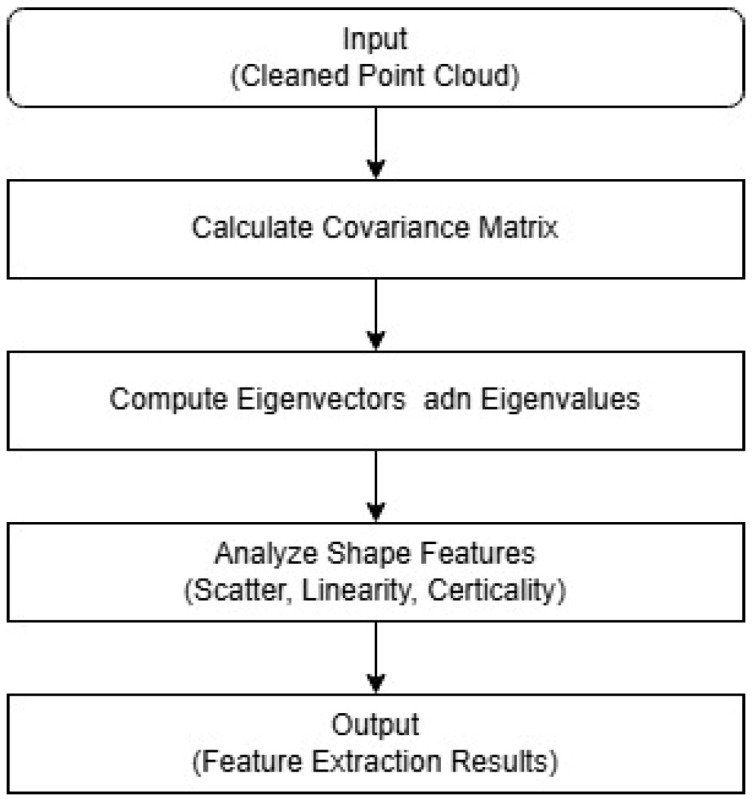
Flowchart of PCA for feature extraction.

**Figure 8 sensors-25-02979-f008:**
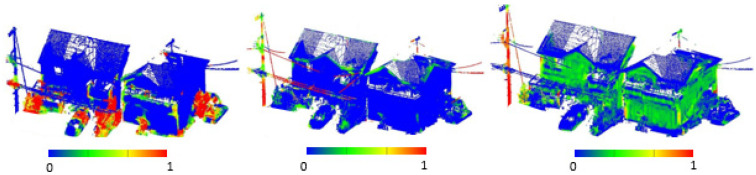
Enhancement of scatter, linearity, and verticality characteristics.

**Figure 9 sensors-25-02979-f009:**
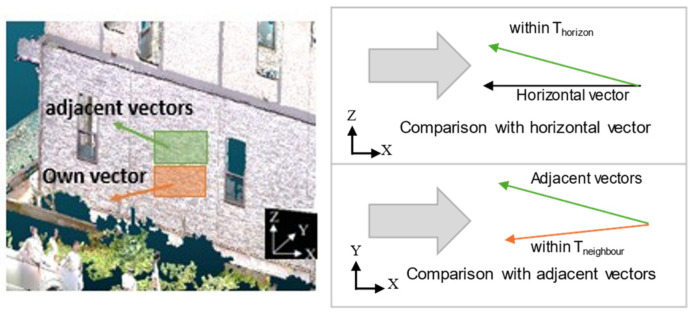
Clustering using the region-growing method.

**Figure 10 sensors-25-02979-f010:**
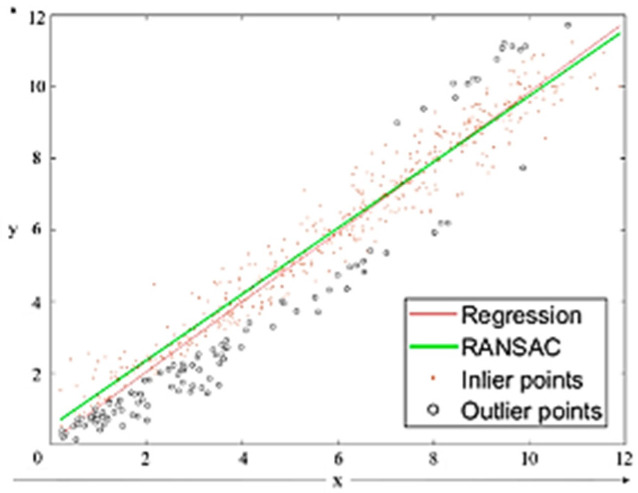
Comparative Results of RANSAC and Traditional Regression Models [15].

**Figure 11 sensors-25-02979-f011:**
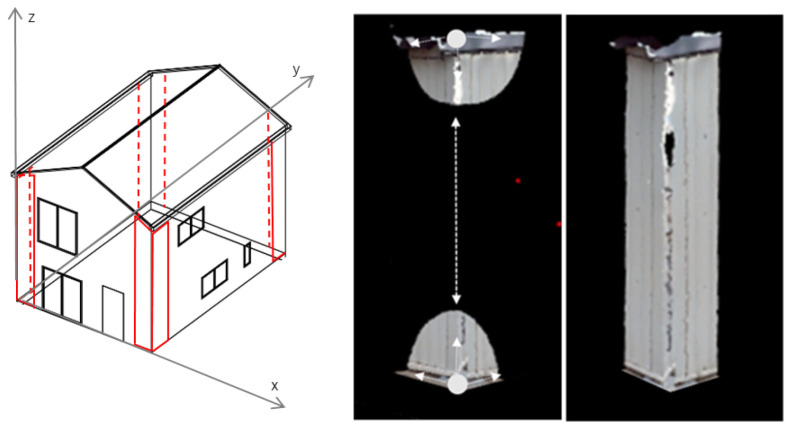
Corner extraction process [15].

**Figure 12 sensors-25-02979-f012:**
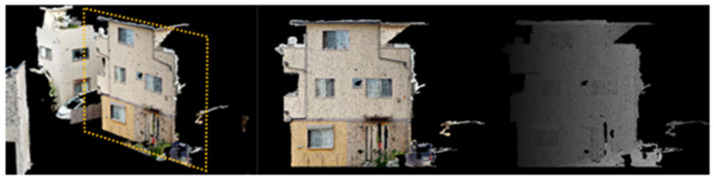
Point cloud projection image and depth image.

**Figure 13 sensors-25-02979-f013:**
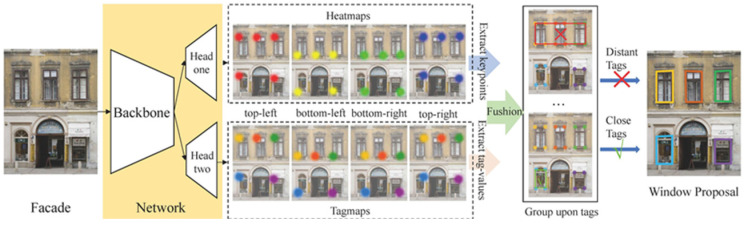
Window Detection iFacades Using Heatmaps Fusion [17].

**Figure 14 sensors-25-02979-f014:**
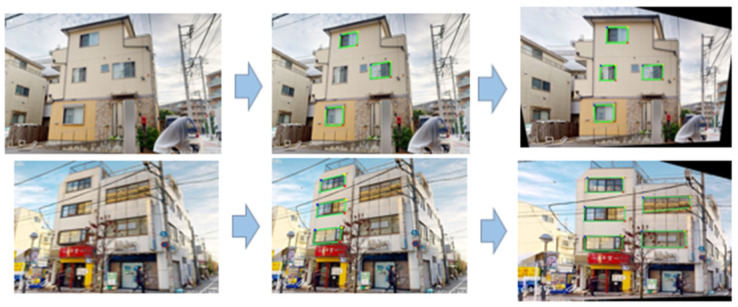
Accuracy differences in window detection before and after frontalization.

**Figure 15 sensors-25-02979-f015:**
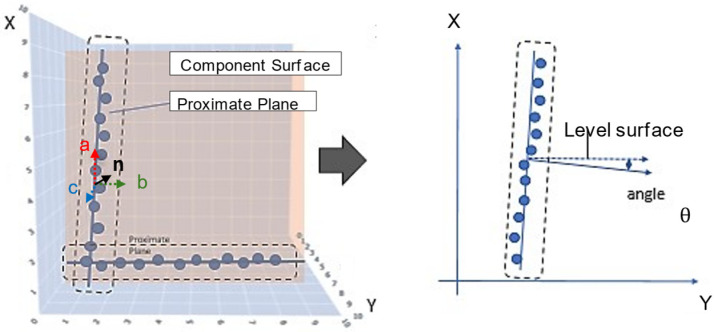
Inclination calculation.

**Figure 16 sensors-25-02979-f016:**
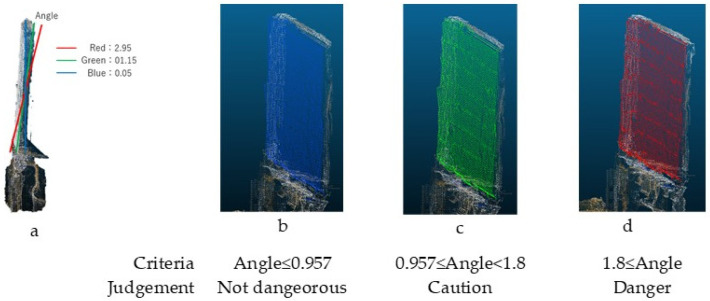
Coloring results according to three angles [15]. (**a**) Illustration of the setup where three inclination angles were input and visualized in red, green, and blue according to the respective judgment criteria. (**b**) Wall section judged as “Not dangerous”, displayed in blue. (**c**) Wall section judged as “Caution”, displayed in green. (**d**) Wall section judged as “Danger”, displayed in red.

**Figure 17 sensors-25-02979-f017:**
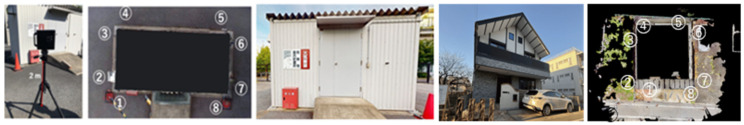
Model of Experiment 1 [15].

**Figure 18 sensors-25-02979-f018:**
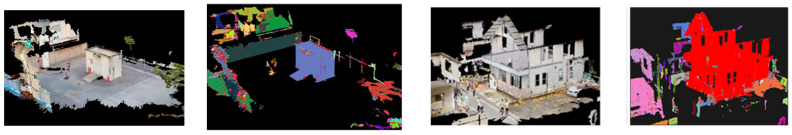
Flow of building extraction process [15].

**Figure 19 sensors-25-02979-f019:**
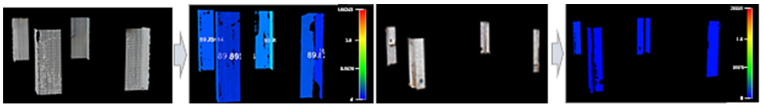
Visualization results of corner measurement.

**Figure 20 sensors-25-02979-f020:**
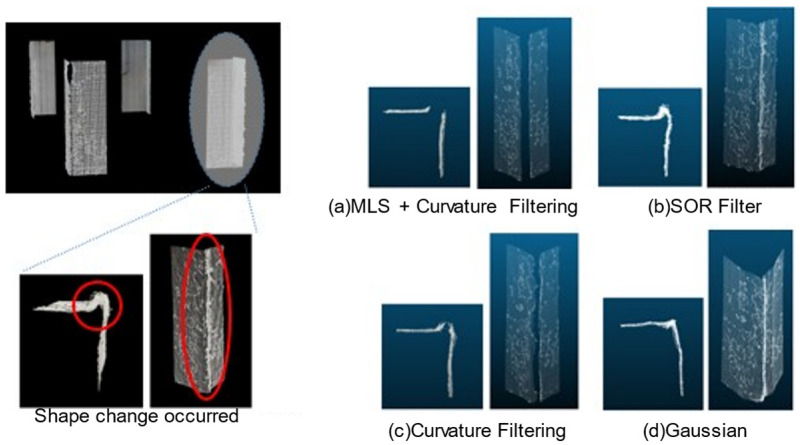
Point cloud at the corner where the shape change occurred [15].

**Figure 21 sensors-25-02979-f021:**
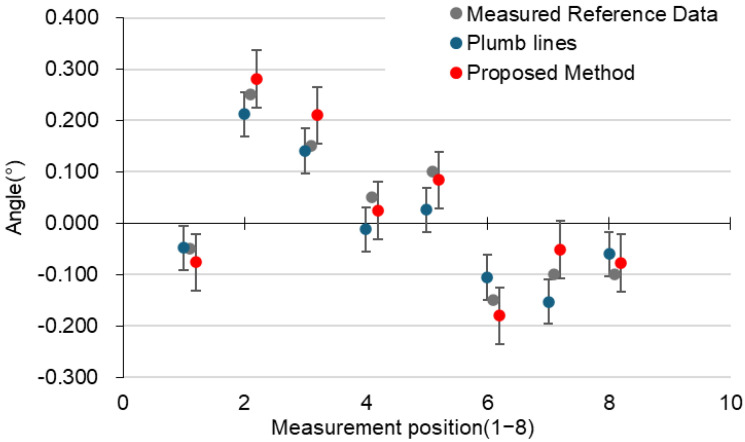
Prefabricated tilt measurement results [15].

**Figure 22 sensors-25-02979-f022:**
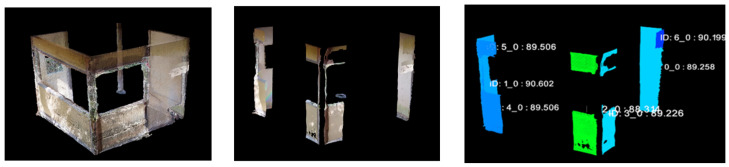
Corner point cloud extraction and angle measurement of the Azumaya.

**Figure 23 sensors-25-02979-f023:**
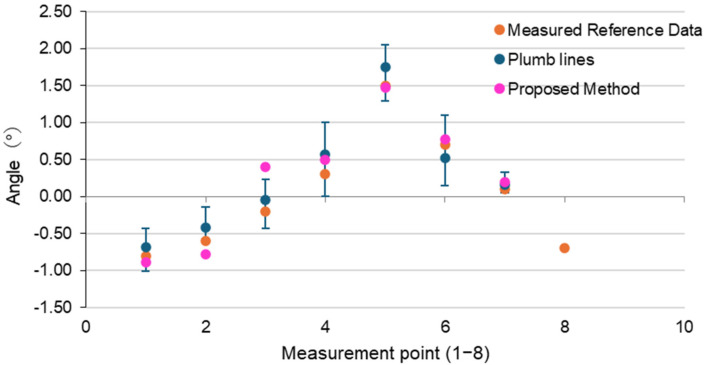
Measurement results of the Azumaya corner section.

**Figure 24 sensors-25-02979-f024:**
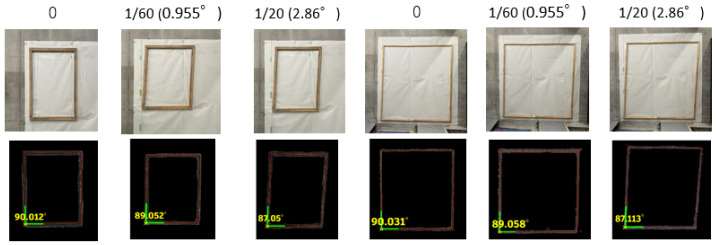
Window frame extraction and angle measurement.

**Figure 25 sensors-25-02979-f025:**
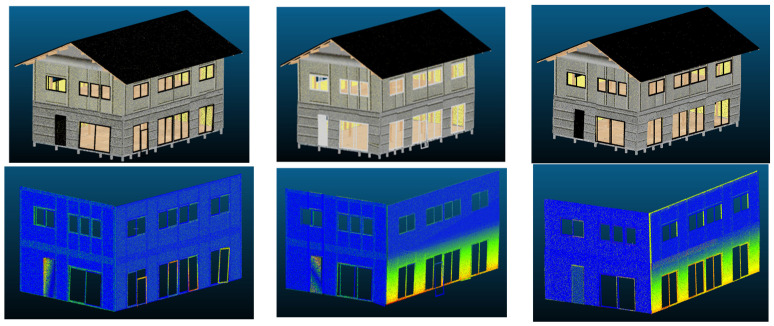
Visualization of inclination results.

**Figure 26 sensors-25-02979-f026:**
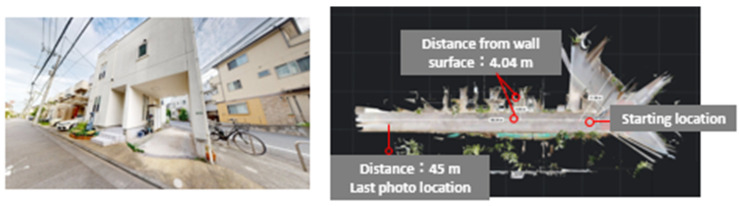
Experimental setup for residential area imaging.

**Figure 27 sensors-25-02979-f027:**
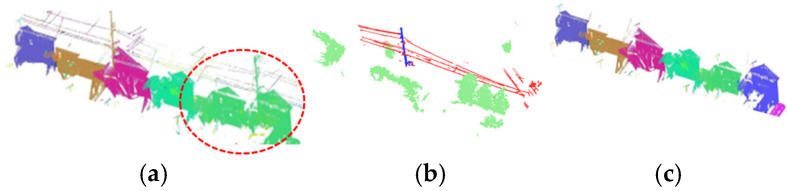
Segmentation results of individual buildings within a dense residential area. (**a**) No filtering process. (**b**) Obstacles detected. (**c**) After filtering.

**Figure 28 sensors-25-02979-f028:**
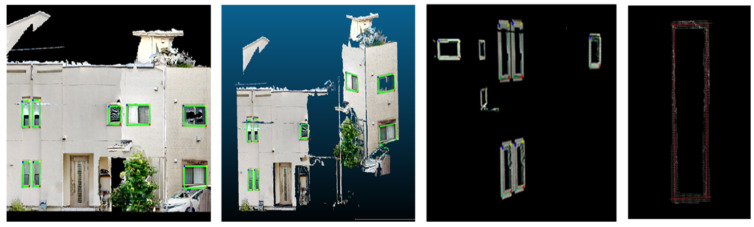
Window frame extraction.

**Table 1 sensors-25-02979-t001:** The overview of the 3D scanning camera [12].

Maker	Matterport, Inc. (Sunnyvale, CA, USA)
Product	Matterport Pro2
Accuracy	Within a 1% error margin for a 10 m (33 ft) distance
Measurement Range	Up to 90 m^2^ (969 ft^2^) per scan

**Table 2 sensors-25-02979-t002:** Comparison of surface properties when each method is applied [15].

Method	Normal Deviation	Mean Error	Standard Deviation of Curvature
MLS + Curvature Filtering	0.55°	0.0495°	0.01
Gaussian Filter	0.93°	0.0921°	0.045
SOR Filter	1.52°	0.185°	0.098
Curvature Filtering	0.65°	0.650°	0.022

**Table 3 sensors-25-02979-t003:** Prefabricated tilt measurement results [15].

Point	Measured Reference Data (°)	Average (°)		Standard Deviation (°)		Mean Error (°)	
		Plumb Bob	Proposed Method	Plumb Bob	Proposed Method	Plumb Bob	Proposed Method
1	−0.05	−0.0477	−0.0764	0.0642	0.0510	0.0023	0.0264
2	0.25	0.212	0.281	0.0696	0.0433	0.0423	0.0312
3	0.15	0.141	0.210	0.0520	0.0499	0.0091	0.0023
4	0.05	−0.0119	0.0249	0.0468	0.0517	0.0619	0.0251
5	0.1	0.0263	0.0841	0.0704	0.0692	0.0737	0.0159
6	−0.15	−0.105	−0.180	0.0455	0.0594	0.045	0.056
7	−0.1	−0.153	−0.0505	0.0700	0.0579	0.0528	0.0495
8	−0.1	−0.0597	−0.0770	0.0306	0.0564	0.0403	0.0597

Note: Red text indicates metrics where the proposed method outperformed the conventional method.

**Table 4 sensors-25-02979-t004:** House tilt measurement results [15].

Measuring Point	Measured Reference Data (°)	Average (°)		Mean Error (°)	
		PLUMB BOB	Proposed Method	Plumb Bob	Proposed Method
1	0.00	89.905	89.997	0.095	0.003
2	0.05	89.876	90.037	0.074	0.087
3	0.10	90.00	89.901	0.1	0.001
4	0.05	89.952	89.973	0.102	0.123
5	0.10	89.905	89.052	0.005	0.052
6	0.05	89.928	89.957	0.022	0.007
7	0.00	89.876	89.907	0.124	0.093
8	0.00	89.905	89.983	0.095	0.017

Note: Red text indicates metrics where the proposed method outperformed the conventional method.

**Table 5 sensors-25-02979-t005:** Azumaya tilt measurement results.

Measuring Point	Measurement Reference Data (°)	Average (°)		Mean Error (°)	
		Plumb Bob	Proposed Method	Plumb Bob	Proposed Method
1	−0.975	−0.680	−0.882	0.115	0.0872
2	−0.596	−0.412	−0.773	0.186	0.177
3	−0.202	−0.051	0.398	0.152	0.600
4	−0.297	0.572	0.494	0.273	0.197
5	1.498	1.750	1.469	0.252	0.029
6	0.355	0.531	0.771	0.175	0.416
7	0.101	0.163	0.199	0.059	0.098

Note: Red text indicates metrics where the proposed method outperformed the conventional method. Underlined values indicate points where the measured inclination exceeded the judgment criteria.

**Table 6 sensors-25-02979-t006:** Window frame measurement results.

Width [W] × Height [H] mm	780 × 970 mm			1690 × 1830 mm		
Angle [rad]	0	1/60 = [0.955°]	1/20 = [2.86°]	0	1/60 = [0.955°]	1/20 = [2.86°]
1st	0	0.95	2.95	0.03	0.942	2.89
2nd	0.05	0.948	2.97	0.08	0.944	2.8
1st	0.03	0.959	2.8	0.01	0.97	2.79
2nd	0.05	0.971	2.78	0.02	0.965	2.78
Depth tilt	0.05	0.07	0.09	0.01	0.09	0.07
MPE	0.442%	0.957%	2.875%	0.340%	0.955%	2.815%

## Data Availability

The original contributions presented in this study are included in the article. Further inquiries can be directed to the corresponding author.

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
