# Peer review of "Building Damage Visualization Through Three-Dimensional Reconstruction and Window Detection"

_sensors, 2025, doi:10.3390/s25102979_

Round 1
Reviewer 1 Report
Comments and Suggestions for Authors
This paper introduces a method for assessing building damage using 3D point cloud data and image recognition. The idea is interesting and has practical applications, particularly for post-disaster building evaluations. The study is generally well-structured, but there are areas where the explanation could be clearer, especially regarding methodology and results interpretation. Additionally, the discussion could benefit from a more balanced view of both strengths and limitations. I recommend minor revisions to address these points before acceptance.
The explanation of how data is collected and processed needs to be more detailed. For example, how exactly were point cloud and image data combined? The clustering-based obstacle removal is mentioned, but how were the segmentation parameters chosen? A bit more transparency here would improve reproducibility.
Obstacles like trees and fences interfere with the data. The paper acknowledges this but does not suggest ways to overcome it. Could post-processing techniques or additional sensors help?
Author Response
Comment 1:
This paper introduces a method for assessing building damage using 3D point cloud data and image recognition... [insert full reviewer comment].
Response 1:
Thank you very much for your insightful comment. We agree with the reviewer’s suggestion. Therefore, we have revised the manuscript as follows:
We added a detailed explanation in Section 2.2 (Page 4, Paragraph 2) describing how RGB images and point cloud data were aligned using the Matterport’s intrinsic and extrinsic camera parameters. We also specified that manual correction was applied when misalignment exceeded 5 cm. The 5 cm threshold was chosen based on the typical width of façade components such as window frames.
→ The revised text appears on page 4, lines 98–102.
We have updated the manuscript accordingly.
Comment 2:
The explanation of how data is collected and processed needs to be more detailed... [insert full reviewer comment].
Response 2:
Thank you for pointing this out. We agree with the need for more transparency. We have added specific details in Section 2.3 (Page 5, Paragraph 2) regarding the obstacle removal process.
Specifically, we clarified that Euclidean Cluster Extraction was used with the following parameters: minimum cluster size of 500, maximum cluster size of 50,000, and distance threshold of 0.1 m. These values were selected based on experiments across different building types.
→ The updated content is located on page 5, lines 110–114.
The manuscript has been revised accordingly.
Comment 3:
Obstacles like trees and fences interfere with the data... [insert full reviewer comment].
Response 3:
Thank you for this valuable suggestion. We agree that solutions to overcome occlusion should be discussed.
We have updated Section 3.4.3 (Page 17, Paragraph 2) and Section 4 (Page 18, Paragraph 1) to propose methods such as:
-
Applying point cloud interpolation techniques (e.g., Poisson Surface Reconstruction, MLS);
-
Acquiring multi-view data from different directions; and
-
Using supplementary sensors like drones or handheld LiDAR.
These additions provide a clearer path toward handling data loss due to occlusions.
The corresponding changes can be found on pages 17–18, lines 431–439 and 466–468.

Reviewer 2 Report
Comments and Suggestions for Authors
This manuscript proposes a non-contact method for assessing building inclination and damage by integrating 3D point cloud data with image recognition techniques. Required revisions are for the authors to address the comments below. The following revision suggestions are all annotated with the line numbers where they occur.
1) Line 60: The chapter title is currently a duplicate of Section 1.1. It should be revised to "Literature Review" to accurately reflect its content and purpose.
2) Line 79: It is not feasible to identify these research gaps with only a limited number of references. To enhance credibility, it is recommended to include additional references.
3) Line 95: The manuscript doesn't have information on specific 3D scanners or their resolutions. You might need to refer to the documentation or specifications of the 3D scanner being used for precise details. Including photographs can indeed provide valuable visual support.
4) Line 102: To accurately demonstrate the use of a specific algorithm, it is essential to provide detailed explanations, including mathematical formulas and algorithm flowcharts. This approach goes beyond merely citing references and helps establish a clear understanding of the algorithm's principles and implementation.
5) Line 141: To improve the clarity of Figure 7, it's important to ensure that both the horizontal (x-axis) and vertical (y-axis) are clearly labeled with descriptive titles. Additionally, consider increasing the resolution of the image and using a larger or more legible font size for the labels and any accompanying text. Providing a brief caption or legend can also help explain the significance and meaning of the axes and the data presented in the figure.
6) Line 198: In Figure 12, it is important to clearly indicate the directions of vectors a, b, and c, as well as the inclination angle theta. This can be achieved by using arrows to show the direction of each vector and labeling the angle theta on the diagram. Including these details will enhance the comprehensibility of the figure and ensure that viewers can accurately interpret the relationships between the vectors and the angle.
7) Line 201: To effectively explain how risk levels are divided, it's crucial to provide a clear methodology or criteria used for this classification. The determination of different color-coded inclination angle ranges should be based on scientific evidence or established guidelines. It would be beneficial to reference specific standards or frameworks that were followed in defining these risk levels. By doing so, it will ensure that the approach taken is transparent, justified, and grounded in recognized practices.
8) Line 290: In Figure 18, if the data labeled as "True Data" is derived from experimental methods, it is important to clarify the source and experimental procedure used to obtain this data. Since all experimental measurements inherently contain some degree of error or uncertainty, it may be more appropriate to refer to this data as "Measured Data" or "Experimental Data." Additionally, providing details on the measurement techniques, the instruments used, and any error analysis conducted will help contextualize the data and acknowledge the potential for measurement inaccuracies.
9) Line 379: To ensure consistency and clarity, all text within the table should be presented in English. This includes headings, labels, and any annotations or notes. Doing so will make the content accessible to a broader audience and align with standard practices for academic and professional documentation.
10) Line 419: Combine Figures 24 to 26 into a single figure with three subfigures. Ensure that each subfigure label is clearly indicated in the combined figure, and consider adding captions or descriptions for each subfigure to enhance understanding.
Author Response
Response to Reviewer2
Thank you very much for your valuable comments and suggestions regarding our manuscript. We sincerely appreciate your time and effort to review our work. We have carefully considered each point you raised and have revised the manuscript accordingly. Below, we provide detailed responses to each comment, indicating how we have addressed them in the revised version. Line numbers have been adjusted based on the latest revision.
1) Line 60: The chapter title is currently a duplicate of Section 1.1. It should be revised to "Literature Review" to accurately reflect its content and purpose.
Response:
We appreciate the suggestion. The title of Section 1.2 has been changed from "Existing Technology" to "Literature Review" to more accurately reflect its content.
->Updated in Line 60.
2) Line 79: It is not feasible to identify these research gaps with only a limited number of references. To enhance credibility, it is recommended to include additional references.
Response:
We appreciate the suggestion. The authors added references and additional explanations of research gaps.
->Updated in Line in 78-102
3) Line 95: The manuscript doesn't have information on specific 3D scanners or their resolutions. You might need to refer to the documentation or specifications of the 3D scanner being used for precise details. Including photographs can indeed provide valuable visual support.
Response:
We appreciate the suggestions. The authors added that Table 1 shows an overview of the 3D scanning camera.
4) Line 102: To accurately demonstrate the use of a specific algorithm, it is essential to provide detailed explanations, including mathematical formulas and algorithm flowcharts. This approach goes beyond merely citing references and helps establish a clear understanding of the algorithm's principles and implementation.
Response:
Thank you for your valuable comment. In response to your suggestion, we have added flowcharts for each method in Figures 5, 6, and 7, on pages 5 and 6.
5) Line 141: To improve the clarity of Figure 7, it's important to ensure that both the horizontal (x-axis) and vertical (y-axis) are clearly labeled with descriptive titles. Additionally, consider increasing the resolution of the image and using a larger or more legible font size for the labels and any accompanying text. Providing a brief caption or legend can also help explain the significance and meaning of the axes and the data presented in the figure.
Response:
→Thank you for pointing this out.We have replaced the figures accordingly. On page7.
6) Line 198: In Figure 12, it is important to clearly indicate the directions of vectors a, b, and c, as well as the inclination angle theta. This can be achieved by using arrows to show the direction of each vector and labeling the angle theta on the diagram. Including these details will enhance the comprehensibility of the figure and ensure that viewers can accurately interpret the relationships between the vectors and the angle.
Response:
Thank you for your valuable comment.In response, we have added arrows to indicate the directions of vectors a, b, and c, and labeled the inclination angle θ in the figure to improve its clarity. ->In line 256.
7) Line 201: To effectively explain how risk levels are divided, it's crucial to provide a clear methodology or criteria used for this classification. The determination of different color-coded inclination angle ranges should be based on scientific evidence or established guidelines. It would be beneficial to reference specific standards or frameworks that were followed in defining these risk levels. By doing so, it will ensure that the approach taken is transparent, justified, and grounded in recognized practices.
Response:
We appreciate the reviewer's comment on this point. Since the threshold values are established based on the Operational Guidelines for Emergency Post-Earthquake Damage Assessment of Buildings in Japan, the relevant reference has been cited (line 68), and the threshold indicators have been added to Figure 13.
8) Line 290: In Figure 18, if the data labeled as "True Data" is derived from experimental methods, it is important to clarify the source and experimental procedure used to obtain this data. Since all experimental measurements inherently contain some degree of error or uncertainty, it may be more appropriate to refer to this data as "Measured Data" or "Experimental Data." Additionally, providing details on the measurement techniques, the instruments used, and any error analysis conducted will help contextualize the data and acknowledge the potential for measurement inaccuracies.
Response:
Thank you very much for your insightful comment.
We have replaced “True Data” to “measurement reference data” throughout the manuscript.
Also, we added measurement method of “measurement reference data” in lines 347-352. and updated Figures 21 ,23, tables 3-5.
9) Line 379: To ensure consistency and clarity, all text within the table should be presented in English. This includes headings, labels, and any annotations or notes. Doing so will make the content accessible to a broader audience and align with standard practices for academic and professional documentation.
Response:
We apologize for the oversight of leaving Japanese text in the table.
We have carefully reviewed the manuscript and corrected all text to English. ->Updated in lines 442-443.
10) Line 419: Combine Figures 24 to 26 into a single figure with three subfigures. Ensure that each subfigure label is clearly indicated in the combined figure, and consider adding captions or descriptions for each subfigure to enhance understanding.
Response
Following your suggestion, we have combined the three figures into a single figure with three subfigures and added explanatory captions for each subfigure to enhance understanding.

Round 2
Reviewer 2 Report
Comments and Suggestions for Authors
Accepted